# Full-Chain FeCl_3_ Catalyzation Is Sufficient to Boost Cellulase Secretion and Cellulosic Ethanol along with Valorized Supercapacitor and Biosorbent Using Desirable Corn Stalk

**DOI:** 10.3390/molecules28052060

**Published:** 2023-02-22

**Authors:** Jingyuan Liu, Xin Zhang, Hao Peng, Tianqi Li, Peng Liu, Hairong Gao, Yanting Wang, Jingfeng Tang, Qiang Li, Zhi Qi, Liangcai Peng, Tao Xia

**Affiliations:** 1Biomass & Bioenergy Research Center, College of Plant Science & Technology, Huazhong Agricultural University, Wuhan 430070, China; 2Key Laboratory of Fermentation Engineering (Ministry of Education), National “111” Center for Cellular Regulation & Molecular Pharmaceutics, Cooperative Innovation Center of Industrial Fermentation (Ministry of Education & Hubei Province), College of Biotechnology & Food Science, Hubei Key Laboratory of Industrial Microbiology, Hubei University of Technology, Wuhan, 430068, China; 3College of Engineering, Huazhong Agricultural University, Wuhan 430070, China; 4College of Life Science, Huazhong Agricultural University, Wuhan 430070, China; 5State Key Laboratory of Herbage & Endemic Crop Biotechnology, School of Life Sciences, Inner Mongolia University, Hohhot 010070, China

**Keywords:** metal catalyst, biomass saccharification, electroconductivity, cellulase secretion, corn stalk

## Abstract

Cellulosic ethanol is regarded as a perfect additive for petrol fuels for global carbon neutralization. As bioethanol conversion requires strong biomass pretreatment and overpriced enzymatic hydrolysis, it is increasingly considered in the exploration of biomass processes with fewer chemicals for cost-effective biofuels and value-added bioproducts. In this study, we performed optimal liquid-hot-water pretreatment (190 °C for 10 min) co-supplied with 4% FeCl_3_ to achieve the near-complete biomass enzymatic saccharification of desirable corn stalk for high bioethanol production, and all the enzyme-undigestible lignocellulose residues were then examined as active biosorbents for high Cd adsorption. Furthermore, by incubating *Trichoderma reesei* with the desired corn stalk co-supplied with 0.05% FeCl_3_ for the secretion of lignocellulose-degradation enzymes in vivo, we examined five secreted enzyme activities elevated by 1.3–3.0-fold in vitro, compared to the control without FeCl_3_ supplementation. After further supplying 1:2 (*w*/*w*) FeCl_3_ into the *T*. *reesei*-undigested lignocellulose residue for the thermal-carbonization process, we generated highly porous carbon with specific electroconductivity raised by 3–12-fold for the supercapacitor. Therefore, this work demonstrates that FeCl_3_ can act as a universal catalyst for the full-chain enhancement of biological, biochemical, and chemical conversions of lignocellulose substrates, providing a green-like strategy for low-cost biofuels and high-value bioproducts.

## 1. Introduction

Lignocellulose is the most renewable biomass resource on Earth; it is convertible into bioethanol and bioproduction for reduced net-carbon release [1,2]. For bioethanol processes, three major steps are principally required, including biomass pretreatment for lignocellulose deconstruction, sequential enzymatic hydrolysis for fermentable sugars and final yeast fermentation for ethanol production [3,4]. However, the intrinsic recalcitrance of lignocellulose leads to highly costly pretreatments and low-efficiency enzymatic saccharification [5], which is unacceptable for large-scale cellulosic ethanol production with potential secondary-waste release into the environment [6]. Hence, the use of green-like technology for cost-effective biofuels and high-value bioproducts remains to be explored.

Lignocellulose recalcitrance is principally formed by distinct cell-wall composition, characteristic wall-polymer features, and dense wall-network interlinks. To reduce recalcitrance, various physical and chemical pretreatments have been broadly conducted with diverse lignocellulose resources [7,8]. Although chemical pretreatments can extract partial wall polymers (hemicellulose, lignin) and alter cellulose features (crystallinity, polymerization) to increase lignocellulose accessibility, they mostly require high temperatures and high concentrations, along with additional processes for chemical recycling. Alternatively, liquid-hot-water pretreatment is applicable as a non-chemical and green-like biomass process [9], but its effectiveness is restricted for largely enhanced biomass enzymatic hydrolysis [10].

For the sequential enzymatic hydrolysis of pretreated lignocellulose residues, mixed cellulases are commonly utilized by incubating *Trichoderma reesei* with a desirable biomass substrate as a carbon source for enzyme secretion [11]. In principle, the *T. reesei*-secreted cellulases are the complexes of three different major enzymes: exoglucanase (CBH), endoglucanase (EG), and β-glucosidase (BG). In particular, advanced biotechnology is increasingly explored to improve the *T. reesei* strain secretion of cellulases and xylanases with high activities [12].

In the full utilization of lignocellulose, the remaining biomass residues obtained from enzymatic hydrolysis and yeast fermentation are considered to generate value-added biochar and nanocarbon via thermal–chemical conversion [13,14]. As biocarbon has a stable structure, large specific surface area, and pore volume, it has been widely employed for environmental remediation and electroconductivity [15,16]. Importantly, iron chemicals, such as K_2_FeO_4_ and FeCl_3_, have been supplied as graphitization agents for synchronous activation with pyrolysis to generate graphitic carbons [17,18,19]. Although iron-based catalysts have been implemented to enhance biochemical and thermal–chemical processes of lignocellulose substrates, most iron chemicals are reported to be inhibitors of biomass enzymatic saccharification and the secretion of fungi by cellulases [20]. However, ferric ions are essential for living organisms; they generally produce high-affinity iron-chelating compounds to scavenge the metal necessary for their metabolism [21]. In particular, the presence of chloride could increase cellulase activity [22].

Corn is a highly photosynthetic-efficient C4 crop that annually produces billions of tons of grain and lignocellulose residues [23]. Using the mature stalks of two distinct corn cultivars (Huatiannuo-3/ZH and Xianyu-1171/ZX), in this study, we performed optimal liquid hot water (LHW) pretreatments that were co-supplied with low dosages of FeCl_3_ as the active catalyst to distinctively extract wall polymers for significantly improved lignocellulose recalcitrance, which led to the achievement of near-complete biomass enzymatic saccharification for high bioethanol production in the desired corn cultivar (ZH) [24]. By collecting all the solid lignocellulose residue obtained from the enzymatic hydrolysis, in this study, we examined its adsorption capacity with Cd as the active biosorbent [25]. Meanwhile, the raw stalk of the desired corn cultivar (ZH) was also used as a carbon source to incubate with the *Trichoderma reesei* for secreting high-activity lignocellulose-degradation enzymes by co-supplying low dosages of FeCl_3_ for *T. reesei* cultivation. After harvesting all the supernatants containing the mixed cellulases and xylanases secreted by the *T. reesei* strain, all the remaining lignocellulose residues were also processed to generate graphitic carbon by a classic thermal–chemical reaction co-supplied with 1:2 (*w*/*w*) FeCl_3_ as the catalyst, and the porous carbon was further examined for electroconductivity [13]. Hence, this study demonstrates the multiple roles of FeCl_3_ as a sensitive catalyst enabling the improvement of two full-chain biomass processes for cost-effective biofuels and highly valuable bioproducts with zero biomass-waste release.

## 2. Results and Discussion

### 2.1. Enhanced Biomass Saccharification of Corn Stalks for Bioethanol Production under Optimal LHW Pretreatment with FeCl_3_ Co-Supplement

As generally described for all the experiments conducted in this study (Appendix A), we initially collected the mature stalks of two corn cultivars (ZH and ZX) with distinctive cell-wall compositions (Table 1). In general, the ZH stalk consisted of significantly lower lignin and cellulose levels than those of the ZX at the *p* < 0.01 level (*n* = 3), with similar hemicellulose contents. In particular, the ZH stalk contained more soluble sugars, which should be directly fermentable for bioethanol production. Next, this study performed various liquid hot water (LHW) pretreatments co-supplied with low dosages of FeCl_3_ as the catalyst to enhance sequential-biomass saccharification by measuring the hexose yield released from enzymatic hydrolysis (Figure 1). During the LHW pretreatment at 190 °C for 10 min, 4% FeCl_3_ co-supplement was effective for enhancing biomass enzymatic saccharification in two corn cultivars, and the highest hexose yields at 70% and 95% (% cellulose) were achieved while the LHW temperature was as high as 190 °C (Figure 1A–C). Although the optimal LHW pretreatment (190 °C for 10 min) led to increases in the hexose yields of 1.7- and 1.8-folds in two corn cultivars relative to their raw samples (without pretreatment), the 4% FeCl_3_ co-supplement increased the hexose yields by as much as 3.0- and 4.4-fold (Figure 1D), suggesting that FeCl_3_ catalysis should play much more enhancement roles for biomass enzymatic saccharification in corn stalks. Consistently, the ZH cultivar displayed significantly higher biomass saccharification than that of the ZX, including both hexoses and total sugar yields (Figure 1E), indicating that the lignocellulose substrate of the ZH stalk has relatively few recalcitrant properties. Furthermore, this study performed yeast fermentation by using all the hexoses released from the enzymatic hydrolysis of the pretreated lignocellulose substrate, and the 4% FeCl_3_ co-supplement caused the highest bioethanol yields, of 9.13% and 11.24% (% dry matter), in the two corn cultivars upon the optimal LHW pretreatments (Figure 1F), which were consistent with their increased hexose yields. In addition, the mass-balance analysis revealed that integrating the LHW pretreatments with low-dosage-FeCl_3_ catalysis was effective for wall polymer (lignin, hemicellulose) extraction, leading to near-complete biomass enzymatic saccharification and higher bioethanol production in the desired corn (ZH) cultivar (Appendix A). Therefore, the optimal LHW pretreatment co-supplied with 4% FeCl_3_ can not only provide a relatively cost-effective and green-like biomass process approach, but it may also demonstrate that FeCl_3_ is an efficient catalyst for lessening lignocellulose recalcitrance.

### 2.2. Improved Lignocellulose Recalcitrance from FeCl_3_ Catalysis in Corn Stalks

To understand why the 4% FeCl_3_ supplement with optimal LHW pretreatment could largely extract wall polymers (lignin, hemicellulose) for significantly enhanced biomass enzymatic saccharification, as described above (Figure 1 and Appendix A), we further conducted the Fourier transform infrared (FT-IR) spectroscopic profiling of pretreated lignocelluloses in two corn stalks (Figure 2A,B). As a result, several typical peaks corresponding for the functional groups (C-O-C [26,27], C-H [28,29,30], C-O [31], O-H [31]) associated with wall-polymer interactions were relatively altered in the LHW-pretreated lignocelluloses, particularly for the FeCl_3_-catalyzed substrates (Appendix A), confirming distinct wall-polymer extraction from the optimal LHW pretreatment co-supplied with FeCl_3_. As a consequence, significantly raised cellulose CrI values were detected in the optimal LHW-pretreated residues with 4% FeCl_3_ catalysis (Figure 2C), which could explain the significantly higher wall-polymer extraction via the disassociation of hydrogen bonds with cellulose microfibrils in the two corn cultivars [32]. Although the optimal LHW pretreatments significantly reduced the cellulose DP values at the *p* < 0.01 level (*n* = 3) compared to their raw materials (without pretreatment), the 4% FeCl_3_ co-supplement led to the lowest cellulose DP values detected in the two corn cultivars (Figure 2D), which may explain why cellulose accessibility was mostly raised in the LHW-pretreated residues with 4% FeCl_3_ catalysis (Figure 2E) [33,34]. Using scan electron microscopy, we further observed much rougher surfaces of pretreated lignocellulose residues, particularly from the optimal LHW pretreatments co-supplied with 4% FeCl_3_ (Appendix A), which was consistent with their remarkably increased cellulose accessibility. As cellulose accessibility is a direct parameter accounting for biomass enzymatic hydrolysis [35], the data obtained in this study demonstrate that 4% FeCl_3_ catalysis is mostly effective at reducing lignocellulose recalcitrance for significantly enhanced biomass saccharification through optimal LHW pretreatment conducted in the two corn cultivars.

### 2.3. Enzyme-Undigestible Lignocellulose as Active Biosorbent for Cd Adsorption

Even though the desired corn cultivar (ZH) showed a near-complete biomass enzymatic saccharification under the optimal LHW pretreatment co-supplied with 4% FeCl_3_, this study collected all the enzyme-undigestible solid-lignocellulose residue to detect its adsorption capacity with Cd using our recently established approach [25,36]. Compared with the residue obtained from the direct enzymatic hydrolysis of the desired corn stalk (without pretreatment), the enzyme-undigestible residue after the optimal LHW pretreatment showed significantly reduced Cd adsorption, by 1.5-fold, at the *p* < 0.01 level (Figure 3). However, the undigestible residue from the optimal LHW pretreatment co-supplied with 4% FeCl_3_ was of the highest Cd adsorption capacity among the three residue samples examined, which was almost 1.9-fold higher than that of the residue from the optimal LHW pretreatment only. Therefore, integrating optimal LHW pretreatment with 4% FeCl_3_ catalysis not only mostly reduced the lignocellulose recalcitrance for the increased biomass enzymatic saccharification, but also its undigestible residue was fully applicable as an active biosorbent for high Cd adsorption without any zero-biomass-waste release.

### 2.4. FeCl_3_ Supplement for Upgraded Lignocellulose-Degradation-Enzyme Activity Secreted by T. reesei Incubation with Corn Stalk

As FeCl_3_ acts as an efficient catalyst for reducing the lignocellulose recalcitrance of the desired corn stalk during the LHW pretreatment performed above, we attempted to test the other role of FeCl_3_ in enhancing the *T. reesei* secretion of lignocellulose-degradation enzymes. Using our recently established approach [37,38], the raw material of the corn ZH stalk was incubated in vivo with *T. reesei* as the carbon source co-supplied with FeCl_3_ at different concentrations (0.01%, 0.05%, 0.1%), and the supernatants were then collected for enzyme-activity assay in vitro (Figure 4). As a comparison with the control (without FeCl_3_), the 0.05% FeCl_3_ supplement led to significantly raised enzyme activities, by 1.3–3.0 fold, for all five assays of the *T. reesei* secretion (Figure 4A–E), indicating that FeCl_3_ can act as an active biological catalyst for fungi secretion from multiple cellulases and xylanase. However, while a high concentration (0.1%) of FeCl_3_ was supplied to the *T. reesei* incubation, three enzyme activities (filter paper, β-glucosidase, xylanase) were deeply inhibited and only CBH activity and EG activity remained slightly high relative to the control. Meanwhile, the FeCl_3_ supplement at the extremely low concentration (0.01%) only significantly raised the CBH activity and xylanase activity, but it also led to higher total protein production (Figure 4F), suggesting that *T. reesei* secretion could be sensitive to FeCl_3_ catalysis during its incubation with lignocellulose substrate. Notably, this study compared these five enzyme-activity assays with the previously reported assays enhanced by supplying Mn^2+^ or Co^2+^ or Ca^2+^ in fungal incubation [39,40,41], and the 0.05% FeCl_3_ supplement enhanced the most enzyme activities examined in this study (Table 2). Based on the performance of SDS-PAGE, this study finally identified a total of 11 enzymes secreted by *T. reesei* incubation with the desired corn stalk co-supplied by 0.05% FeCl_3_ (Table 3, Appendix A), which confirms the production of various cellulases and xylanases at high levels of activity secreted by *T. reesei*.

### 2.5. Porosity-Raised Biocarbon Generated by FeCl_3_ Activation with T. reesei-Undigested Lignocellulose Residues

Given that the *T. reesei* incubation co-supplied with 0.05% FeCl_3_ led to high-activity enzyme secretion, as described above, a large amount of undigested lignocellulose residues remained. Hence, in this study, we further generated porous carbons by performing classic thermal–chemical conversion co-supplied with FeCl_3_ at a proportion of 1:2 (*w*/*w*) into the *T. reesei*-undigested residue substrate of the desired corn stalk (Figure 5). Based on the BET assay by N_2_ absorption, this study observed a typical sorption isotherm accountable for the presence of micro-pores in the carbon of the control *T. reesei*-undigested residue (without 1:2 FeCl_3_ activation), whereas a small hysteresis loop under the medium-pressure range was found for the meso-pores in the *T. reesei*-undigestible residue activated by 1:2 FeCl_3_ (Figure 5A), suggesting a distinct pore-size distribution of biocarbon activated by 1:2 FeCl_3_ [42]. Furthermore, we found that the 1:2 FeCl_3_ activation raised the porous volume much more than, and the specific surface area of the biocarbon twofold, compared with the control sample (Figure 5B,C), suggesting that FeCl_3_ activation plays an enhancement role in the generation of highly porous biocarbon [18]. During the TEM observation, the FeCl_3_-activated sample showed graphene-like carbon with fewer layers than that of the control sample (Figure 5D), which was consistent with its significantly increased porosity. The Raman scanning revealed a typical G-peak (~1580 cm^−1^), which represented the graphene-like carbon generated in the FeCl_3_-activated sample (Figure 5E) [43], and the XRD assay further confirmed two diffraction peaks at 24° and 44°, which were slightly shifted in relation to the diffraction peaks (2θ) for the (002) and (100) planes of the graphite (Figure 5F) [44,45]. However, the lack of a 2D peak in the Raman scanning suggested a relatively poor quality of graphene-like carbon. Furthermore, in this study, we applied XPS to detect the element composition and chemical characteristics of the FeCl_3_-activated sample (Figure 5G–I). As a result, the FeCl_3_-activated sample mainly contained carbon (C) at 88.01% and oxygen (O) at 11.78% with tiny iron (Fe) at 0.14%, suggesting that the 1:2 FeCl_3_ supplement mainly acted as a chemical catalyst for the generation of highly porous carbon (Figure 5G). Individual peaks were accordingly identified, such as sp^2^-bonded carbon (284.5 eV), sp^3^-C (285.0 eV), C-O (286.2 eV), C=O (288.1 eV) and O-C=O (288.9 eV) (Figure 5H) [46,47]. In addition, the XPS O 1s spectrum showed two major peaks corresponding to C=O (532.1 eV) and C-O (533.6 eV) (Figure 5I) [48], which may have served as the active sites for the high-porosity biocarbon generated in this study.

### 2.6. Improved Supercapacitor Performance of the Porous Carbon Generated by FeCl_3_ Activation

Since the highly porous carbon was generated by 1:2 FeCl_3_ activation with the *T.* reesei-undigested lignocellulose residues, as described above, this study tested its electrochemical performance using our previously established approaches [45]. In general, three types of carbon sample all displayed quasi-rectangular shapes, which represented the good character of the double-layer capacitor (Figure 6A and Appendix A). However, during a comparison with the controls of the two carbon samples (without FeCl_3_ for thermal–chemical conversion), the FeCl_3_-activated carbon showed a higher current response at different scan rates for better capacitive properties, which were mainly due to its larger specific surface area and porous volume, as described above. Galvanostatic charge–discharge (GCD) tests were also conducted with three carbon samples, and a typical inverted “V” shape was observed, suggesting a classic capacitive response and fast charge transfer (Figure 6B). Consistently, the FeCl_3_-activated carbon sample displayed a much longer discharging time for the highest specific capacitance in the electrolyte systems examined among the three carbon samples. Furthermore, the specific capacitances were evaluated by the discharge times at different current densities, and the FeCl_3_-activated carbon sample showed much higher specific capacitances than those of the two control samples, by 3–12-fold (Figure 6C), which confirmed that the 1:2 FeCl_3_ supplement effectively catalyzed the thermal–chemical conversion of the *T.* reesei-undigested lignocellulose residues into highly porous carbon as the supercapacitor. Nevertheless, the specific capacitances of the FeCl_3_-activated carbon sample remained lower than the carbons generated from other biomass resources, according to previous reports [17,49,50,51], indicating that optimal FeCl_3_ activation should be explored in future studies. In addition, of the two controls, the carbon sample generated from the *T.* reesei-undigested residues co-supplied with 0.05% FeCl_3_ also displayed consistently higher electroconductivity than that of the carbon sample without the 0.05% FeCl_3_ supplement, which suggests that the 0.05% FeCl_3_ supply was also effective at improving the production of *T.* reesei-incubated lignocellulose residue for the generation of highly porous carbon.

## 3. Materials and Methods

### 3.1. Collection of Mature Stalks in Two Corn Cultivars

Two corn cultivars (Huatiannuo-3/ZH and Xianyu-1171/ZX) were grown in the Experimental Field of Huazhong Agricultural University. The mature stalks of two corn cultivars were collected, dried at 50 °C, and ground into the powders through a 40-mesh screen. The soluble sugars of corn stalks were extracted with potassium-phosphate buffer (0.5 M, pH 7.0); the solid residues were rinsed with deionized water until reaching pH 7.0, dried, and stored in a dry container until use.

### 3.2. Wall-Polymer Extraction and Determination

Wall-polymer extraction and assay were conducted as previously described [52], with minor modifications [53]. Total lignin was measured by the Laboratory Analytical Procedure of the National Renewable Energy Laboratory with minor modification [54]. All assays were completed in independent triplicates.

### 3.3. Detection of Wall-Polymer Features

The degree of polymerization (DP) and crystalline index (CrI) of cellulose samples were detected as previously described [55]. Cellulose accessibility was estimated by performing Congo red (CR) staining as described in [56], with minor modifications [55]. Monosaccharides of hemicellulose were analyzed by GC-MS (SHIMADZU GCMS-QP2010 Plus, Berlin, Germany), as described in [57].

### 3.4. Lignocellulose Observation and Characterization

Biomass morphology was observed under scanning electron microscopy (SEM, Gemini 500, Obercohen, Germany), as previously described [8]. Lignocellulose chemical linkages were detected by Fourier transform infrared (FT-IR) method (Thermo Fisher Scientific, Waltham, MA, USA), as described in [58].

### 3.5. Liquid Hot Water (LHW) Pretreatment Co-Supplied with FeCl_3_

The biomass samples (0.300 g) were mixed with 2.4 mL FeCl_3_ at various concentrations (0.5%, 1%, 2%, 4% and 6% *w*/*v*). The well-mixed samples were placed into stainless-steel bomb with PTFE jars and a thermostatic magnetic stirrer (Kerui Instrument Co., Ltd., Gongyi, China) under shaking at 60 rpm. The samples were incubated at 150 °C, 170 °C, 190 °C and 210 °C for 5 min, 10 min, 15 min and 20 min, respectively. After reactions, the bombs were cooled to room temperature, the pretreated samples were rinsed several times with deionized water until pH 7.0, and the remaining solid residues were collected for enzymatic hydrolysis. All assays were conducted in independent triplicates.

### 3.6. Enzymatic Hydrolysis and Yeast Fermentation for Bioethanol Production

The pretreated lignocellulose residues were rinsed with potassium-phosphate buffer (0.2 M, pH 4.8) and then incubated under 5% (*w*/*v*) solid loading with 6 mL (2.0 g/L) of mixed cellulases (HSB, containing cellulases at 13.25 FPU/g biomass and xylanase at 8.40 U /g biomass from Imperial Jade Bio-technology Co., Ltd., Yinchuan, China) for 48 h under shaking at 150 rpm, at 50 °C, and co-supplied with 1% (*v*/*v*) Tween-80. After centrifugation at 4000× *g* for 5 min, the supernatants were collected for hexose and pentose assay, and the solid residuals were collected for Cd-adsorption assay.

Yeast fermentation and ethanol measurement were conducted as previously described [59]. About 0.5 g/L of *Saccharomyces cerevisiae* (Angel Yeast Co., Ltd., Yichang, China) was incubated with the supernatants collected from enzymatic hydrolyses of pretreated lignocellulose residues, and the fermentation was conducted at 37 °C for 48 h. Ethanol was measured using the K_2_Cr_2_O_7_ method [60]. All assays were conducted in independent triplicates.

### 3.7. Cd-Adsorption Analysis

The solid residues remaining from enzymatic hydrolysis were washed with ultra-pure water and dried as biosorbent for Cd-adsorption analysis, as previously described [36]. Solution of 5 mg/L Cd^2+^ was prepared by adding Cd(NO_3_)_2_∙4H_2_O into ultra-pure water. The adsorption experiment with 0.025 g biosorbent was performed in 50-milliliter tubes for 4 h under shaking at 150 rpm. After the adsorption experiment, the samples were filtered through a 0.45-micrometer-membrane filter, and the residual Cd^2+^ concentration in the filtrate was detected by flame atomic adsorption spectrophotometer (FAAS HITACHI Z-2000, Tokyo, Japan) equipped with air-acetylene flame, as previously described [36]. The Cd adsorption at equilibrium q_e_ (mg/g) and the percentage removal efficiency (%R) were estimated as described in [61].

### 3.8. T. reesei Strain Cultivation Induced by FeCl_3_ and Enzyme-Activity Determination

The *T. reesei* Rut-C30 strain (CICC 40348, from China Center of Industrial Culture Collection) was grown on potato-dextrose agar (PDA) at 30 °C for 7 d, and the spores were harvested with double distilled water and counted on the hemocytometer. The germination rate of spores was accurately examined for appropriate incubation time prior to micro-fluidic analysis and sorting. The spores were collected and adjusted to a density of 1 × 10^7^ spores/mL. A spore suspension of about 500 μL was added into 30 mL of liquid cellulase-inducing medium, incubated at 30 °C under shaking at 200 rpm for 7 d. The liquid cellulase-inducing medium was prepared as previously described [37]. About 0.6 g of biomass powder of corn stalk without soluble sugars was added into the liquid cellulase-inducing medium as carbon source and induced by FeCl_3_ at various concentrations (0%, 0.01%, 0.05%, and 1%; *w*/*v*).

Filter-paper activity (FPA) of crude cellulase solutions secreted by *T. reesei* was determined as described previously [37]. Exoglucanase (CBH), endoglucanase (EG), β-glucosidase (BG), and xylanase activities of crude cellulase solutions secreted by *T. reesei* were examined in vitro by using carboxymethylcellulose sodium (CMC-Na), 4-Nitrophenyl β-D-cellobioside (pNPC), salicin, and beechwood xylan (from China National Pharmaceutical Group Co., Ltd., Shanghai Yuanye Bio-Technology Co., Ltd., Shanghai, China) substrates dissolved with sodium-citrate buffer (0.05 M, pH 4.8) as previously described [37,62]. All assays were completed in independent triplicates.

Total proteins secreted by *T. reesei* were estimated as previously described [37]. The SDS-PAGE was run for profiling all proteins secreted by *T. reesei* using stain-free precast gels (Zoman Biotechnology Co., Ltd., Beijing, China), according to the manufacturer’s instructions. All crude enzymes secreted by *T. reesei* were analyzed by LC–MS/MS (Jingjie PTM BioLab Co., Ltd., Hangzhou, China; Orbitrap Elite LC–MS/MS, Thermo, Waltham, MA, USA), as described in [37]. Liquid-chromatography–MS/MS-analysis data were identified by searching the *T. reesei* Rut-C30 protein-sequence databases downloaded from Uniprot (https://www.uniprot.org/), accessed on 30 April 2022.

### 3.9. Preparation of Biocarbon Materials

The remaining solid residue obtained from *T. reesei* cultivation with corn (ZH) stalk was mixed with FeCl_3_ (1:2, *w*/*w*), and ground into the powder. The powder sample was loaded into a tube furnace (OTF-1200X-60UV, Kejing Material Technology Co., Ltd., Hefei, China), heated under N_2_ at a rate of 5 °C/min up to 1000 °C for 2 h, and then cooled down to 300 °C at a rate of 10 °C/min. After cooling to room temperature, the excess ferric ions from the samples were rinsed with 1 M of HCl aqueous solution for 6 h. Next, the carbon residues were rinsed with deionized water until reaching pH 7.0, and the remaining residues were treated by ethanol under ultrasound for 2 h in a sonicator, and dried as biocarbon materials.

### 3.10. Characterization of Porous Carbons

The biocarbon materials were observed under transmission electron microscope (TEM, JEM-2800, Tokyo, Japan) and analyzed by X-ray diffraction (XRD, Advance D8) and Raman spectrum (Thermo Scientific DXR, Waltham, MA, USA). The elements and binding energy of carbon materials were detected by X-ray photoelectron spectroscopy (XPS, Thermo Scientific K-Alpha, Waltham, MA, USA). Specific surface area and pore diameter were analyzed by Automated Surface Area and Porosity Analyzer (Micromeritics ASAP 2460, Norcross, GA, USA)

### 3.11. Measurement of Electrochemical Properties

The electrochemical properties of biocarbon materials were evaluated as previously described [45]. About 0.032 g of biocarbon materials, 0.004 g of Super-P, and 0.004 g of PTFE were mixed with ethanol as the dispersant. Biocarbon materials and PTFE acted as the active materials and binder, respectively. The working electrode was prepared by pressing the above mixture onto a current collector, which was nickel foam (1 cm × 1 cm). The test was performed in 6 M of KOH with a three-electrode system, in which a Hg/HgO and a platinum plate were applied as the reference and counter electrodes, respectively.

Cyclic voltammetry (CV) and galvanostatic charge–discharge (GCD) were performed on a CHI660E electrochemical workstation to evaluate the electrochemical performance. The CV was performed at scan rates of 100 mV/s, and GCD was carried out at current densities from 0.5–20 A/g. The specific capacitance (C, F/g) based on GCD test was calculated by the following formula: C = I × Δt/(m × ΔV). In this formula, I, Δt and ΔV represent for the discharging current (A), discharging time (s) and discharging voltage (V) excluding the IR drop during the discharging process; m (g) is the mass of active material in the working electrode [63].

## 4. Conclusions

By performing optimal LHW pretreatment with corn stalks, this study found that 4% FeCl_3_ co-supplement led to near-complete biomass saccharification for high levels of bioethanol, and all the enzyme-undigestible residues were applicable as active biosorbents for Cd adsorption. When incubating the corn stalks with *T. reesei* for the secretion of lignocellulose-degradation enzymes, the 0.05% FeCl_3_ co-supplement increased the secreted-enzyme activities by 1.3–3.0-fold, and the undigested residue was further activated by the 1:2 FeCl_3_ to generate highly porous and graphene-like carbon, which was applicable as a supercapacitor with specific capacitances raised by 3–12-fold. Hence, this study demonstrated that FeCl_3_ is a universal catalyst for low-cost bioethanol and high-value bioproduction with zero biomass-waste release.

## Figures and Tables

**Figure 1 molecules-28-02060-f001:**
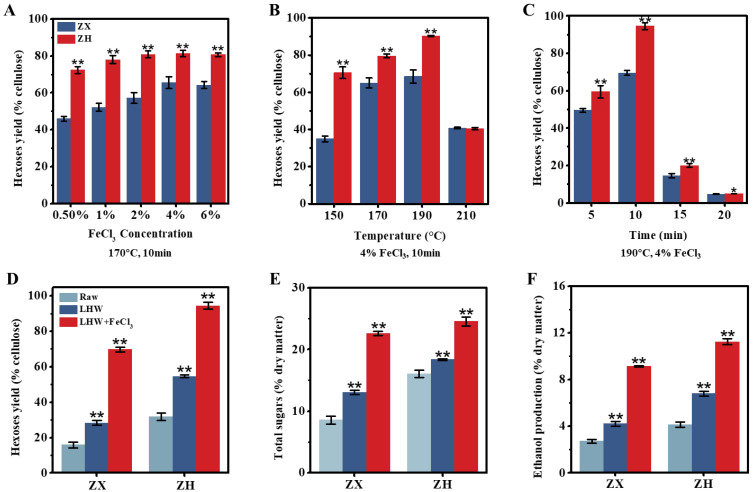
Optimal liquid hot water (LHW) pretreatment co-supplied with FeCl_3_ for biomass enzymatic saccharification and bioethanol production in two corn (ZX, ZH) cultivars. (**A**–**C**) The LHW pretreatments under different conditions by measuring hexose yields released from enzymatic hydrolyses of pretreated residues; (**D**,**E**) hexose and total sugar yields under two pretreatments (optimal LHW at 190 °C for 10 min with/without 4% FeCl_3_) relative to the control (raw material without pretreatment), total sugars calculated from all hexoses and pentoses released from enzymatic hydrolysis; (**F**) ethanol yield obtained from yeast fermentation by using all hexoses as carbon sources. Data as mean ± SD (*n* = 3); * or ** a significant difference between two corn cultivars (**A**–**C**) or pretreated residue and raw material at *p* < 0.05 or 0.01 level.

**Figure 2 molecules-28-02060-f002:**
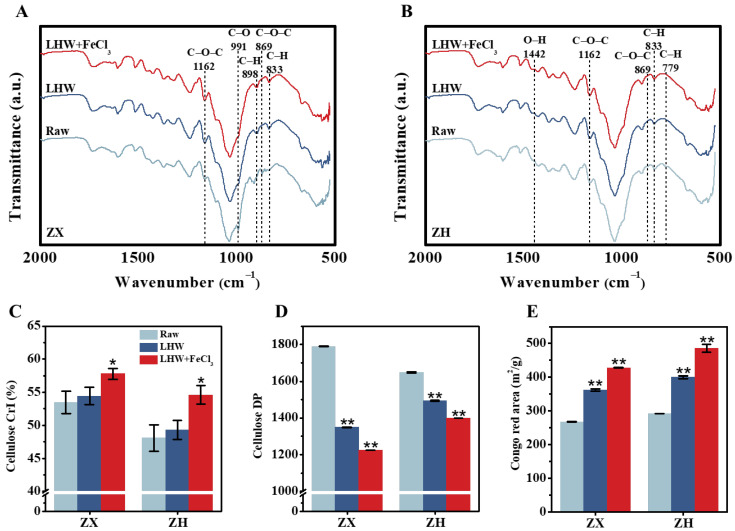
Characterization of wall-polymer linkages and cellulose features after optimal LHW pretreatment co-supplied with 4% FeCl3 in two corn cultivars. (**A**,**B**) Fourier transform infrared (FT-IR) spectroscopic profiling, as annotated in Appendix A; (**C**,**D**) cellulose CrI and DP; (**E**) cellulose accessibility by Congo red staining. Data as mean ± SD (*n* = 3); * or ** a significant difference between pretreated residue and raw material at *p* < 0.05 or 0.01 level.

**Figure 3 molecules-28-02060-f003:**
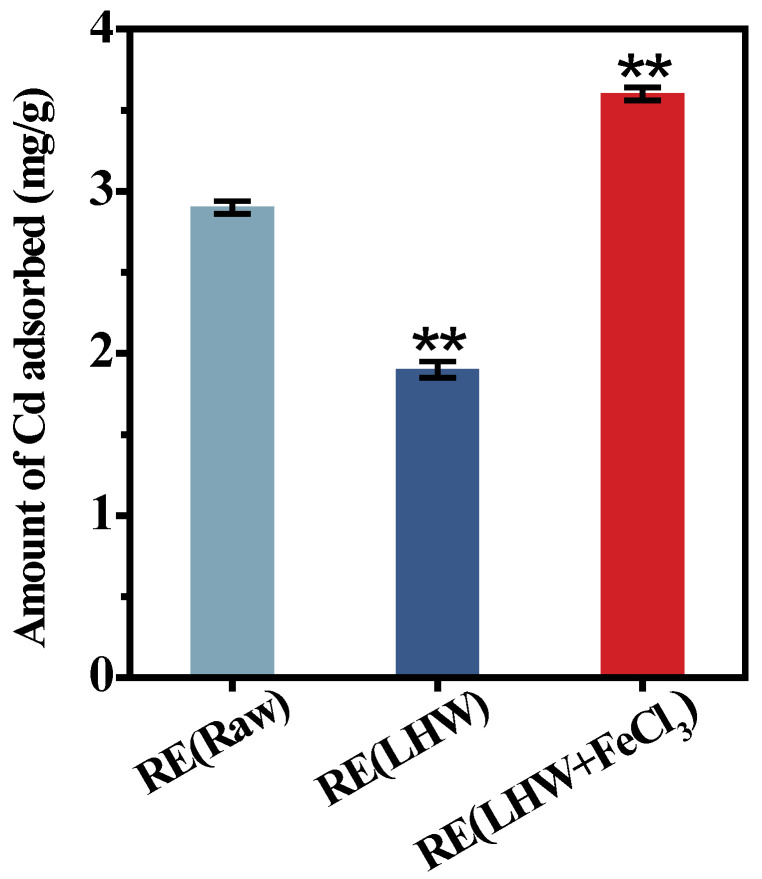
The Cd-adsorption capacity with the enzyme-undigestible residues as biosorbent upon the optimal LHW pretreatment co-supplied with/without 4% FeCl_3_ in the desirable corn (ZH) cultivar. Data as mean ± SD (*n* = 3); ** a significant difference between pretreated residues and raw material at *p* < 0.01 level.

**Figure 4 molecules-28-02060-f004:**
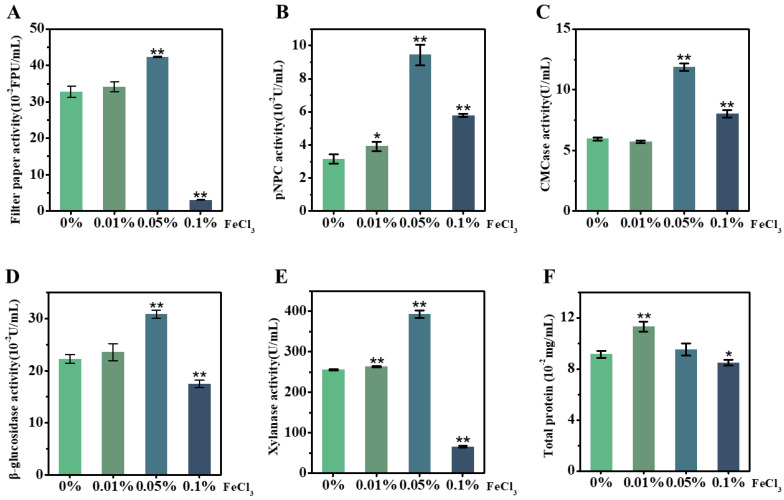
Activity assays in vitro of lignocellulose-degradation enzymes secreted by *T. reesei* incubation with raw stalk of ZH cultivar co-supplied with different concentrations of FeCl_3_ in vivo. (**A**) Filter-paper activity; (**B**) CBH; (**C**) EG; (**D**) BG; (**E**) xylanase; (**F**) total protein of supernatant. Data as mean ± SD (*n* = 3); * or ** a significant difference between FeCl_3_-supplied samples and control (without FeCl_3_) at *p* < 0.05 or 0.01 level.

**Figure 5 molecules-28-02060-f005:**
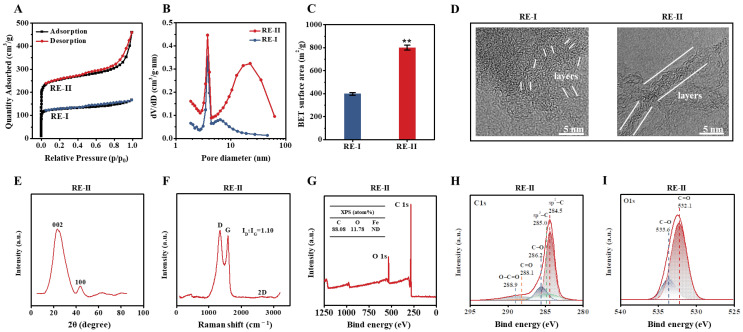
Characterization of the porous carbons generated by 1:2 (*w*/*w*) FeCl_3_-activated thermal–chemical conversion of the lignocellulose residues (RE) from *T. reesei* incubation with raw ZH stalk co-supplied with 0.05% FeCl_3_. (**A**–**C**) For the two carbon generated from RE-I (Raw + 0.05% FeCl_3_) and RE-II (Raw + 0.05% FeCl_3_) + 1:2 FeCl_3_: (**A**) TEM observation; (**B**) nitrogen adsorption/desorption isotherm profiling; (**C**) pore-size-distribution profiling; (**D**) BET surface-area assay. (**E**–**I**) For the carbon sample generated from RE-II: (**E**) X-ray assay; (**F**) Raman-spectra profiling; (**G**) XPS-spectra profiling; (**H**,**I**) high-resolution XPS spectrum at C1s region and high-resolution XPS spectrum at O1s region (**I**). ** a significant difference between two lignocellulose residues at *p* < 0.05 or 0.01 level.

**Figure 6 molecules-28-02060-f006:**
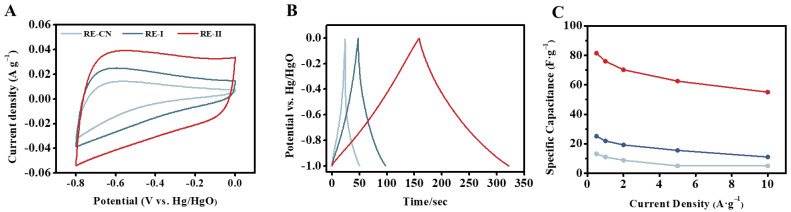
Electrochemical properties of the porous carbon generated by 1:2 (*w*/*w*) FeCl_3_-activated thermal–chemical conversion of the lignocellulose residues (RE) from *T. reesei* incubation with raw ZH stalk co-supplied with 0.05% FeCl_3_. (**A**–**C**) For the three carbon samples generated from RE-I (Raw + 0.05% FeCl_3_), RE-II (Raw + 0.05% FeCl_3_) + 1:2 FeCl_3_ and RE-CN (raw, incubated with *T. reesei* without FeCl_3_): (**A**) Cyclic-voltammetry curves at 100 mV s^−1^; (**B**) galvanostatic charge/discharge curves at 0.5 A/g; (**C**) specific capacitance calculated by galvanostatic charge–discharge at different current densities.

**Table 1 molecules-28-02060-t001:** Cell-wall compositions (% dry matter) of two corn (ZX, ZH) cultivars.

Samples	Wall Polymer Level (% Dry Matter)			
	Cellulose	Hemicellulose	Lignin	Soluble Sugar
ZX	32.63 ± 1.49	17.82 ± 0.06	16.7 ± 0.38	17.77 ± 0.38
ZH	26.37 ± 0.60 **	19.30 ± 0.26 **	13.05 ± 0.63 **	22.87 ± 0.94 **

** A significant difference between ZX and ZH samples at *p* < 0.01 level (*n* = 3).

**Table 2 molecules-28-02060-t002:** Comparison of enzyme activities secreted by fungi co-supplied with various metals.

Inducing Substrate	Increased Enzymatic Activities (%)					Reference
	FPA	CBH	EG	BG	Xylanase	
Fe^3+^	29.38	199.37	99.16	38.38	53.72	This study
Mn^2+^	- *	94	62	-	-	[40]
Co^2+^	-	-	58.62	-	-	[41]
Ca^2+^	-	40	80	-	40	[39]

* The data are not available.

**Table 3 molecules-28-02060-t003:** The LC–MS/MS assay of *T. reesei*-secreted enzymes incubated with corn stalk and 0.05% FeCl_3_.

Protein Name	Accession No.	iBAQ (×10^6^)	MW [kDa]
Exoglucanase I	P62694	3009	54.07
Endo-β-1,4-glucanase I	G0RKH9	656.77	48.21
Endo-β-1,4-glucanase II	G0RB67	44.16	254.04
Endo-β-1,4-glucanase VII	A0A024SFJ2	53.24	26.8
β-D-glucosidase	Q12715	50.95	78.43
β-glucosidase	G0RDY1	3.42	84.68
Endo-1,4-β-xylanase	A0A1L7H884	4144.40	20.77
Endo-1,4-β-xylanase I	P36218	189.98	24.58
Xyloglucanase	A0A024S9Z6	192.46	87.13
β-xylosidase	Q92458	238.57	87.19
β-xylanase	Q9P973	143.56	38.08

## Data Availability

Not applicable.

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
