# Peer review of "Full-Chain FeCl3 Catalyzation Is Sufficient to Boost Cellulase Secretion and Cellulosic Ethanol along with Valorized Supercapacitor and Biosorbent Using Desirable Corn Stalk"

_molecules, 2023, doi:10.3390/molecules28052060_

Round 1
Reviewer 1 Report
the manuscript is well structured and also far too extensive in my opinion, it would probably have been better to break it into two parts, one focusing on degradative performance and the other on the potential applications of non-fermentable products such as lignin, starting from 2.5 of the results, in my opinion, it could definitely be another manuscript.
Anyway, here are my comments:
Major
do not use abbreviations in the abstract
the authors should determine or provide some bibliographic evidence, that the FeCl3 concentrations used are not toxic and do not cause cell lysis
verifying that it is actually proteins that are secreted that are found, and in the case of secretion by what mechanism they are exported
it would be interesting to have an explanation as to why even if the amount of protein in 0.01% FeCl3 is higher (slightly), the degradative activities are better with 0.05% FeCl3
Minor:
Line 29 T.reesei
check that the name T. reesei is in italics throughout the manuscript
Table 2 add * close all the data not available or detectable
Reviewer 2 Report
This paper is well-written and the reading is easy. Some parts of this work have been already published but some novelty was added and all these new data bring useful information.
Reviewer 3 Report
This manuscript should be accepted after major revision, my comments are below:
1. Abstract and title of Manuscript should be changed. It is not totally clear the aims and objectives of paper.
2. Raman spectra need to be deconvoluted.
3. XPS spectra of Fe must be deconvoluted and discussed.
4. Please, explain what is battery-suprcapacitors? It can not be both.
5. CV at different scan rates must be provided.
6. Capacitance retention at different scan rates must also be provided.
7. GCD at different current densities must be provided.
Round 2
Reviewer 1 Report
I thank the authors for taking my comments into consideration, the manuscript is much improved. My only perplexity remains on table 2, where despite being asked, there are values that are not shown, it is not clear whether they are not available or have been omitted. I suggest improving this point
Author Response
Thanks for kind comments and the concern with the unpresented data of Table 2. We carefully checked the orignial articles, and could not find the data that are available. So we revised the Table 2 “The data are not available.”
Reviewer 3 Report
It can be accepted in present form.
Author Response
Thanks for encouraging comments.